# Regulating Citywide Inclusive Sanitation (CWIS) in Colombia

**DOI:** 10.3390/ijerph19095669

**Published:** 2022-05-06

**Authors:** Analía Saker, Andrea Bernal Pedraza, Abishek Sankara Narayan

**Affiliations:** 1Aguaconsult, Colchester CO7 9GS, UK; analiasaker123@gmail.com; 2Centro de Pensamiento en Cultura, Territorio y Gestión—Universidad Nacional de Colombia, Manizales 170017, Colombia; andreabernalpedraza@gmail.com; 3Eawag—Swiss Federal Institute of Aquatic Science, 8600 Dübendorf, Switzerland; 4ETH—Swiss Federal Institute of Technology, 8092 Zürich, Switzerland

**Keywords:** Citywide Inclusive Sanitation, regulatory framework, urban sanitation, Colombia

## Abstract

The conventional top-down scope of relying only on centralised sewerage has proven insufficient to reach the entire global population with safely managed sanitation and meet Sustainable Development Goals 6.2. and 6.3 by 2030. Citywide Inclusive Sanitation (CWIS) has emerged as an approach to accelerate progress by considering different technologies and service provision models within the same city to expand sanitation access equitably and sustainably. However, to generate an enabling environment for CWIS to be implemented successfully, regulatory frameworks must be adapted, as they are often unsuited for non-sewered sanitation solutions. By analysing the Colombian case study through a mixed qualitative methodology comprised of a policy review, semi-structured interviews, and workshops with key stakeholders in the urban sanitation sector, the country’s regulatory framework was evaluated to determine if it is adequate to implement CWIS. Regulations were identified to pose barriers for CWIS and produced a disabling environment for its application. This research proposes recommendations to adapt the regulatory framework to allow CWIS application in Colombia based on the encountered barriers. This is the first comprehensive study on regulations for CWIS in the Latin American context and therefore provides the basis for further research to understand the dynamics related to effective regulations for CWIS globally.

## 1. Introduction

Appropriate sanitation is essential to preserve health and environmental well-being, and to allow economic progress. Nevertheless, 3.6 billion people currently use sanitation services that leave human waste untreated, and 494 million people practise open defecation [1]. The lack of proper sanitation has detrimental human health effects caused by poor water quality, adverse environmental impacts due to the degradation of water systems, and negative economic consequences on account of the ecosystem’s productivity losses and public health expenses [2,3,4]. In addition, poor sanitation negatively affects gender inequality, women’s safety and school attendance [5]. When evaluating the progress towards universal sanitation, urban settings stand better than rural settings regarding basic access to sanitation (85% urban vs. 59% rural). However, when safely managed sanitation is compared between urban and rural, the difference is marginal (47% urban vs. 42% rural) [6]. Hence, effective actions to ensure universal, safely managed sanitation in urban settings are still needed, especially given that the pressure on public services is expected to increase as urban areas continue to rapidly urbanise, with a projected annual rate of 1.73% from 2020 to 2025 [7].

Since the 1960s, urban sanitation provision has been tackled as a top-down, resource-intensive approach favouring almost exclusively sewerage networks with centralised water treatment and a single service provider [8]. Even though this strategy has served High-Income Countries (HICs) to achieve nationwide coverage, it has fallen short in achieving sanitation for all in Low and Middle-Income Countries (LMICs) [9]. In this sense, LMICs face challenges that do not exist in HICs. They suffer from rapid and unplanned urbanisation, a lack of resources to finance infrastructure, and weak institutions to enforce regulations [10]. These characteristics in LMICs make reaching the most vulnerable zones in urban contexts difficult with the conventional approach to sanitation.

Sustainable Development Goals (SDG) 6.2 and 6.3 set the target of reaching everyone with safely managed sanitation by 2030. However, the current sanitation progress rate is too slow and would need to quadruple to achieve the goals by 2030 [10]. Aware of the need for accelerating progress towards safely managed urban sanitation, in 2017, experts in the sanitation sector proposed a paradigm shift [8,11]. Through sectoral consensus, Citywide Inclusive Sanitation (CWIS) was introduced as a novel approach to urban sanitation in which all the city residents (including the most vulnerable) have equitable access to adequate, affordable, and improved sanitation services considering various technological options (sewered and non-sewered) and service provision models throughout the whole sanitation service chain [11]. The fundamental principles for CWIS are prioritising the human right to sanitation, managing human waste in an environmentally friendly manner, considering an array of technological alternatives, and planning in an inclusive and holistic way with all the stakeholders, where authorities operate with a clear and inclusive mandate that considers a mix of business models [11]. Thus, CWIS promotes equity by acknowledging the sanitation challenges in LMICs and reaching the most vulnerable through a differentiated approach that tailors solutions adequate to the local conditions. 

CWIS has gained traction among development actors, academics, and practitioners during the past five years. Until now, research in CWIS has been majorly based on planning methodologies to choose appropriate technologies and service provision models [12,13], documentation of successful case studies [14,15], and the development of theoretical frameworks [16,17]. As this is a novel approach that proposes a paradigm shift, research on different fronts is required to integrate this approach within the countries’ policies effectively. In this regard, several authors have stressed the importance of the *enabling environment* for CWIS [18,19,20]. The enabling environment is defined as the broader structural and institutional context that frames sanitation service delivery. It has five fundamental pillars: sufficient budgeting and financial mechanisms, coherent planning and monitoring, required knowledge and skills, proper institutional arrangements, and adequate policy and strategy [16]. To implement CWIS, the enabling environment must change in a comprehensive manner as all the pillars are interlinked and depend on the others. Several authors have documented experiences where appropriate-to-the-context technologies failed to succeed due to the absence of an enabling environment to ensure financial viability, sustainability, and accountability [21,22,23,24]. 

Within the adequate policy and strategy aspect of the enabling environment, a suitable regulatory framework has been recognised as an essential element for CWIS [25,26]. The Eastern and Southern African Water and Sanitation Regulators Association (ESAWAS) has led the research on this front [27]. They analysed member countries’ regulatory frameworks’ preparedness to incorporate the CWIS approach through a GAP analysis and proposed a series of guidance documents on accountability, planning and institutional roles for CWIS [19,28]. On the other hand, some countries like Japan [29], Malaysia [14], and India [23,30] have recently included in their regulatory frameworks certain aspects that align with the CWIS approach by incorporating diverse service delivery models and technological alternatives for urban sanitation. Even though some progress has been made in understanding how to adjust regulations for CWIS, and there is momentum around on this topic research is still incipient and, in most cases, based on regulating only one technology type. Furthermore, studies in this area have primarily been in African, Southeast Asian and South Asian countries. For South and Central American countries facing different challenges, this topic has been seldomly explored. In Latin American countries, literature about urban sanitation is scarce, and it is still focused on regulating the conventional urban sanitation approach [31,32,33]. Considering that most South and Central American countries need to accelerate progress to reach the SDGs, new research is required to adapt the regulatory framework to a more inclusive approach to urban sanitation. 

To address the gap in the literature, this research focuses on the Colombian case study and analyses if the regulatory framework is appropriate to implement CWIS. Colombia has already covered most of the urban population with basic sanitation (92.5%), has dramatically improved basic sanitation coverage from 80% to 92% from 2000 to 2020 and almost eliminated open defecation in this same period. However, advancement in sanitation coverage has reached a plateau, where the progress rate has significantly decreased compared to the 90s and will not be sufficient to meet the SDGs 6.2 and 6.3 by 2030 [34]. Additionally, Colombia has made little progress in safely managed sanitation and has one of the lowest coverage rates (16.6%) among South American countries [1]. Concerned by the slow progress in urban sanitation in Colombia, government representatives have shown the political will to tackle urban sanitation challenges by creating differentiated service provision schemes for underserved areas in complex contexts [35]. Therefore, Colombia’s case is interesting to analyse as the country has been willing to adapt its policy towards sanitation and can give valuable insights on what has worked in terms of regulations and what is missing to adopt CWIS. 

This research aims to identify the challenges and areas for development in the Colombian regulatory framework to adopt CWIS successfully. It also seeks to generate significant insights to guide policies in other countries in the region facing similar challenges as well as to assess the drivers for the development or lack thereof of regulatory mechanisms [36]. However, this study does not aim to propose a regulatory framework, nor does it identify specific measures to be taken in this regard; it centres on understanding the aspects within the regulatory framework that require further development for CWIS. 

## 2. Methods

This research employed a descriptive case study approach using a qualitative methodology to evaluate the appropriateness of the current regulatory framework to embrace the CWIS approach in Colombia. A case study methodology is applicable for this purpose as a regulatory framework is a social instrument based on the needs of a specific society, and it is related to and determined by the context [37]. Therefore, using a case study methodology allows delving further into the Colombian context while also contributing to a broader understanding of the enabling environment for CWIS that can be applied to other countries. 

Two methods were applied to achieve the aims of the research. Firstly, an in-depth policy review was conducted to analyse the existing regulatory framework. The laws, decrees, resolutions and court rulings related to sanitation were identified through relevant Colombian governmental institutions’ websites. Secondly, semi-structured interviews and workshops were held to analyse the different points of view of the key stakeholders involved in designing, monitoring, and implementing the regulatory framework in Colombia’s water and sanitation sector at a national and a local level. Workshops and interviews served to complement the information obtained from the policy review, explore the institutions’ perspectives, and identify the commonalities and differences in perception among the stakeholders. Interviewees were asked about the challenges of reaching SDG targets for sanitation, their knowledge about CWIS, its applicability to the Colombian context, and the suitability of the Colombian current regulatory framework to incorporate this new approach (the interview protocol is provided in the Appendix A). 

The initial sampling for the interviews was purposeful through stakeholder mapping and then complimented with snowballing. Seventeen in-depth interviews and four workshops were conducted online with 30 participants (Table 1). Informed consent was obtained via email, and it was orally confirmed at the beginning of each interview. The interviews were conducted in Spanish and transcribed verbatim. They were analysed and coded in NVIVO in Spanish, and the findings were translated to English.

For the scope of this study, the interviewees were selected from key institutions with responsibility in the WSS sector, utilities experienced in complying with regulations, and experts in the Colombian WSS regulatory framework. As this research exclusively aimed at identifying the barriers to CWIS in the regulatory framework in Colombia, representatives of the community were not included. However, further research into the construction of regulatory frameworks must consider the community viewpoint to guarantee that sanitation services are affordable, acceptable, and equitable for end users.

A gap analysis was used to evaluate the sanitation situation under the current regulatory system and propose recommendations to implement CWIS. A gap analysis is a methodology used to determine the missing steps to move from a current state to a desired future state [19]. In addition, the case study of Tumaco city CWIS pilot was evaluated under the lens of the PESTLE (political, economic, social, technological, legal, and environmental) model. Academics use this business analysis tool to decide about a proposed intervention considering the effects of the changing environment to ensure sustainable diffusion and adoption of the intervention [38,39]. The PESTLE analysis was used to understand the regulatory barriers at a local level and to evaluate the differences between the local and the national level. 

## 3. Results

### 3.1. Institutional Framework of the WASH Sector in Colombia

Colombia is a presidential nation where policy is dictated from the national level and executed by the local authorities. Responsibilities are split between several institutions in the water supply and sanitation (WSS) sector (See Figure 1). The Ministry of Housing, City, and Territory (MHCT) is the head of the sector and dictates the policy with technical support from the National Planning Department (NPD). The Ministry of the Environment and Sustainable Development (MESD) oversees the environmental policy related to water pollution and establishes the standards for water discharge and water abstraction. The Ministry of Agriculture and Rural Development (MARD) is responsible for orienting policy related to the re-use of sanitation by-products for agriculture [40]. The Water Regulatory Commission (WRC) is an independent agency in charge of setting standards and licencing public utilities, and the Superintendence of Public Utilities (SPU) enforces economic and technical regulations by monitoring service providers’ performance [41]. Finally, Autonomous Environmental Corporations (AEC) are decentralised entities that enforce environmental water standards. At the local level, the departmental water plans (DWP) collaborate with the Local Governments (LG) to invest in infrastructure and technical assistance to public utilities.

In Colombia, urban water and sanitation provision is devolved, where LG are responsible for providing water and sanitation to all buildings that abide by the technical construction parameters and are located within the urban perimeter. Likewise, they must obtain funding to carry out the initiatives and choose a service delivery model that suits their context. Law 142 of 1994 opened the door for private businesses, public–private partnerships, and municipally controlled corporatized public utilities to provide services. Regardless of the service delivery arrangement, public utilities must meet service quality and continuity standards, as well as continual coverage expansion, equitable service, and economic efficiency [42].

In the WSS sector, LG have various ways of procuring funding (Figure 2). The General Royalties System (GRS) and the National General Budget (NGB) are national funds for which the LGs compete. To access those resources, LG must submit project proposals to the MHCT, where a technical team approves or rejects the project based on compliance with technical, legal and land ordering requirements according to Res 330-2017 and Res 799-2021. Those projects are usually contested, and many are discarded for not complying with the requisites stated in the Res 661-2019, especially those formulated by weaker LG that lack the technical capacity to develop robust proposals [43]. On the other hand, the General Contributions System (GCS) allocates 5.4% of its resources for subsidies and investments in water and sanitation infrastructure. In Colombia, water and sanitation tariffs are cross-subsidised, wealthier households (Stratum 6-5-4) subsidise poorer households (Stratum 1-2-3). In poorer municipalities, where the tariff alone does not cover the subsidies of poorer households, LG support service providers with resources from the GCS. These resources are readily available for LG at their discretion. Additionally, LG can also generate resources on their own through municipal taxes. However, poorer municipalities usually depend solely on national resources as they do not have the capacity of raising taxes independently. During the 2014–2016 period, the National Planning Department (NPD) reported that the WSS’ financial distribution was: 43% GCS, 12% GRS, 10% NGB, 35% tariffs [34], showing that the WSS is highly dependent on the GCS to fund its water and sanitation initiatives, making it a crucial financial source in implementing CWIS.

There has been some progress in Colombia’s policy approach to incorporate aspects of CWIS into the national water and sanitation strategy. For the technology aspect of CWIS, in 2014, the Colombian government released a policy recommendation document to tackle water and sanitation in rural zones that suggested different technologies for challenging contexts that could not implement conventional solutions [41]. To apply those recommendations, the government introduced an article in the National Planning Law of 2015 that allowed alternative technologies for water and sanitation provision. This article also introduced the concept of *differentiated schemes*, both for rural and special zones within urban areas. Since then, the government has issued a series of regulations to implement the new approach, especially in service provision and technical aspects. As part of this policy, the government launched the *Agua al Barrio* programme to extend water and sanitation services to 203,000 people living in informal settlements by 2022 [44]. The government is still developing regulatory instruments to adapt institutions to incentivise the implementation of this new policy.

### 3.2. Understanding Regulatory Barriers through a Policy Review

In the broad sense, regulations are defined as the sustained and focused control exercised by a public agency over activities valued by a community, which involves setting rules and ensuring their enforcement. For WSS service provision, usually, regulations refer to economic regulations that include “setting, monitoring, enforcement and change in the allowed tariffs and service standards for utilities” [45]. Nevertheless, for the scope of this study, various aspects of regulations will be touched upon as CWIS goes beyond service delivery and intertwines with land management, pro-poor, financial, and environmental regulations. 

As a result of the policy review, eight laws, nine decrees and eight resolutions were related to urban sanitation provision. As shown in Table 2, the sector is mainly governed by Law 142 of 1994, which sets the responsibilities and principles for delivering public services. The regulatory instruments are organised in order of hierarchy. The Constitution serves as the foundation for all other regulations. Following the Constitution in order of precedence come the laws enacted by Congress, the decrees, and, finally, the resolutions. Regulatory instruments were divided into five categories to conceptualise the different aspects involved in sanitation provision: service delivery, land management, environmental, financial, and technical. 

Furthermore, based on the information compiled from the policy review, a gap analysis was conducted for the urban regulatory arrangements through the sanitation service chain. Table 3 shows how the regulatory framework fails to incorporate the CWIS principles within the service delivery scheme. For example, operation and maintenance of onsite sanitation systems are done through contracts between users and private operators without regulated tariffs, standards or indicators to monitor adequate performance. The gap analysis also highlights specific regulatory aspects that require further development to provide an enabling environment for CWIS.

### 3.3. Understanding Regulatory Barriers through Expert Views

Even though all participants were asked the same questions, the answers and perspectives varied widely between individuals and types of stakeholders. The participants only agreed on a few points and, in some cases, had opposing views to their colleagues. At a national level, most of the participants stressed that regulations needed to change, at least in some way, to facilitate CWIS implementation. Table 4 summarises the barriers that regulations pose to CWIS. 

The only aspect where there was general agreement was the desirability of amending Law 142-1994 to give a legal basis for new technologies and new service delivery models for WSS. Participants agreed that the current definition for sanitation (which is:themunicipal collection of residues, mostly liquids, through pipes and channels, and also the complementary activities of transportation, and final disposal of such residues) in the law poses restrictions to implement CWIS, leaving out non-sewered and alternative service delivery models. However, there was no agreement on the level of urgency to carry out this amendment. Four participants stated it was politically problematic because amending this law would change many aspects that would later have to be regulated and require immense effort and coordination to modify this law successfully. Moreover, four participants believed that the country did not have the political atmosphere to implement a reform of this size, which requires congressional approval. For them, it was better to manoeuvre with the current regulatory framework to implement alternative schemes.

As answers, priorities, and perceptions varied greatly among participants, the data gathered was divided into categories, following the same structure as the policy review. The regulatory barriers were divided into service delivery, land management, financial and technical. 

Service delivery

The government introduced an alternative mandate to service delivery in Decree 1272-2017. As a result, the WRA published Res 699-2021, which contained a new scheme that targeted underserved areas, allowing service providers to calculate the tariffs and measure the continuity and quality indicators under more flexible conditions. Despite welcoming the new regulation as a first step to reaching the most vulnerable zones, some interviewees believed that the new regulatory framework would be problematic in its implementation for the following reasons. 

Firstly, Res 699-2021 is still limited to what is stated in Law 142-1994, which means that solutions are restricted to sewered infrastructure, leaving out onsite and shared alternatives. Secondly, the new regulation states that technical parameters can be relaxed only while the challenging context justifies a *provisional scheme*. A provisional scheme is defined as an alternative solution applied only in situations where conventional solutions cannot be implemented. When that provisional scheme period ends because the context is not under those difficult conditions anymore, the infrastructure parameters must again comply with technical regulations for conventional schemes. This is a disincentive for public utilities to invest in alternative solutions because they would have to replace the non-conventional technology at the end of the provisional period, having to invest twice. This makes non-conventional projects financially unattractive to public utilities. Thirdly, the new resolution is 59 pages long and too complex for community or small service providers to understand and apply. The tariff calculation is complicated, and even knowledgeable public servants found the methodology hard to comprehend. Fourthly, participants believed the new framework does not offer an incentive for service providers to expand their coverage to these areas as they do not have an obligation to reach non formalized zones. The regulation lacks a proper scheme to incentivize public utilities to cover these low-paying-capacity areas that require costly infrastructure. Finally, to implement this scheme, the service provider must have a signed commitment from the municipality that states that the zone has a prospect of urbanization. This process is a barrier as it is not always within the mayor’s interest to formalize zones, and it can be problematic to fulfil this requirement. In general, participants believed this regulation improves the flexibility in some respects but still responds to a conventional model and is limited to what Law 142-1994 states. Hence, it is unlikely that it will promote CWIS implementation.

Moreover, people living in these contexts usually do not have the payment capacity to cover the investment and maintenance costs and require a subsidy. Nevertheless, subsidies are only applicable for sewered service provision schemes, so communities that rely on non-sewered technologies are not eligible for financial assistance to cover the maintenance and operation of the infrastructure. 

Land Management

At a national level, representatives of various entities had different interpretations of what the legislation states in land management matters. Three participants questioned the legality of providing public services in zones that are not recognised in the land planning instrument issued by the municipality. Contrary to that, five interviewees stated that Law 2044-2020 not only allowed but obliged local authorities to solve the vulnerability condition of communities in informal settlements by providing services or relocating the people to a safe place. On the financial side, any WSS project that seeks funding from the national government requires a certification of the land legality, a document that states that this area has an urbanization prospect or a document that certifies the land possession. Therefore, if the property legality is not resolved, there is a financial barrier to access national funds to finance projects. Moreover, LG are responsible for updating land management instruments to include peri-urban and marginalized zones within the urban perimeter. Currently, 88% of these instruments are outdated or inaccurate [46]. Participants stated that marginalized communities are left outside planning tools in many cases because majors do not have the political will to urbanise certain zones. Hence, the service delivery in informal settlements is highly dependent on the political intentions of the major with those informal areas.

Interviewees also mentioned, citing Law 388-1997, that in high-risk zones, the state cannot reach with services and should instead focus on reallocating the communities somewhere else. Nevertheless, current regulations do not consider a temporary solution to assist these zones while communities are effectively reallocated. This process usually takes a long time, and while it happens, households are left alone to implement solutions without any regulation that often are not dignified nor appropriate. Finally, participants stated that there is no clarity regarding the legality of implementing development projects in high-risk zones, not even within the government itself. Therefore, public utilities and institutions prefer not to tend to these zones as they might risk facing control entities for breaking urban development regulations.

Environmental

Environmental regulations, according to several interviewees, impede CWIS implementation. The strict permits required by the AECs to develop any sanitation infrastructure were the most frequently mentioned barriers. According to the participants’ points of view, those permits render projects unviable because, in some cases, the expense of obtaining them exceeds the cost of the sanitation facility itself. Regulations for environmental protection are overly stringent, even in circumstances where the environmental impact would be negligible. The AECs, for example, require a ground discharge permit septic tanks. These permits involve water quality models and soil characterisation, and their approval can take more than a year. Participants emphasised that while some service provision standards have developed to meet current challenges, environmental restrictions have remained unchanged. 

Financial

Interviewees mentioned that the regulatory framework posed financial barriers to CWIS. Firstly, they agreed that resources to finance non-conventional infrastructure are limited. The GCS resources cannot be employed on non-conventional solutions, and this is the source on which the water and sanitation sector relies the most (as highlighted in Section 3.1). In this regard, one participant stressed:


*For example, if municipalities want to invest in those areas where sewerage networks cannot be installed, they cannot use GCS resources, even if they have enough GCS funds, because the law does not allow it. So, that is the problem: there is a disincentive for investment. If municipalities wish to invest in septic systems to extend coverage with alternative solutions, they can do so, but it has to be done with other sources of resources.*


Moreover, social costs essential to planning participatory processes and incentivising the adoption of the sanitation solution by the communities cannot be included within the project’s budget. Thus, restricting its designation and risking the overall sustainability and acceptability of the project by the community. 

Technical

Participants had varied opinions regarding technical regulations and their suitability for CWIS. Whilst MHCT’s officials stated that regulations do not exclude any technology, participants from other institutions believed that even though technical regulations do not forbid non-conventional technologies, they are not included nor proposed as possible solutions for urban sanitation. In fact, the Water and Sanitation Technical Regulations Manual (WSTRM) does not even mention faecal sludge management. In this regard, one participant stated: “*whatever the WSTRM says, they [referring to the ministry] are interested in doing centralised solutions because it’s easier to demonstrate progress in the population covered*”. 

### 3.4. Understanding Regulatory Barriers at a Local Level from Tumaco’s Case Study

The case of Tumaco city is particularly relevant since it has several context-related difficulties, such as an inadequate topography, a highly densified city, difficult environmental conditions that could suppose barriers to the application of conventional sewered solutions, and also demonstrates how the municipalities implement the policies and have a different perspective about the barriers to reaching 100% urban sanitation. The national government is planning a CWIS pilot by evaluating a condominial sewerage as an alternative to a centralised sewered approach, which would be too costly and complex to execute in a municipality like Tumaco. In this sense, it can be a valuable learning experience for the government based on lessons learnt in the field that could incentivise regulatory development. 

The Tumaco pilot exposed a context with a series of challenges preventing the effective implementation of WSS service provision programmes. Through a PESTLE analysis (contained in the Appendix A), it was possible to identify the obstacles that need to be tackled before the pilot starts its execution. 

The participants believed that the regulatory framework is not prepared to respond to the particular challenges of the regions. They stressed that Tumaco city had functioned so far by bending or ignoring some regulations. This generates vulnerability to the project, as the control entities can block projects even after their implementation starts. All the interviewees agreed that in order to carry out the project, they would have to exert pressure on the MHCT to relax some requirements because otherwise, the project would be bureaucratically and financially unviable. Specifically, for the condominial sewerage that is being planned, the most significant barrier was the *easement process* required, where each household must agree to the installation of pipelines in their properties. Until now, regulations state that a civil legal procedure is necessary for each house, making the pilot administratively unviable, as there are more than 2000 houses included. 

Further, participants also noted that environmental regulations must be relaxed to start the implementation of the project. Interviewees stated that even for those cases in which a new technology is being piloted, like in Tumaco, with the condominial sewerages, environmental standards remain unchanged, which discourages innovation. Some participants share the view that the disconnection and lack of coordination between the regulations dictated by the MHCT and the MESD are because both Ministries are not pursuing the same objective. Hence, they proposed that a coordination space must be organised with the MESD, MHCT, and the implementing partners to discuss the possibilities of more flexible requirements for the treatment plant effluent standards.

Despite the project’s problems, the difficulties experienced in implementing this pilot have already prompted the government to begin adjusting the regulatory framework. The Ministry updated technical regulations and guidelines as a result of the Tumaco pilot to allow the implementation of condominial sewerages.

### 3.5. Factors That Impede the Development of the Sanitation’s Regulatory Framework

Participants also shared their perspectives about the factors that prevented the development of a sanitation regulatory framework with a more inclusive approach. Firstly, the lack of priority in sanitation was mentioned by some participants. The government prioritises its efforts in water projects over sanitation, as water supply projects are cheaper to implement and more popular among the electorate. A government official in charge of coordinating the MHCT regulatory agenda stated that the institution was not prioritising sanitation and that there were no development initiatives on this topic. 

Moreover, resistance to technological change was widely mentioned during the interviews. Participants noted that public servants responsible for developing and issuing regulations have been in their position for many years and only have experience with conventional solutions, so they do not trust any option different from what they already know. Interviewees described public officials in some institutions in the WSS sector as *close-minded* and *pragmatists* tackling water and sanitation issues with an *engineering lens* only. When asking the WRA experts that crafted the new service provision regulation, they emphasised that “*It has been difficult to break paradigms*”. Some participants mentioned that the resistance to change is partly because only the conventional approach is being taught at universities, so professionals do not conceive any other solutions, nor would they know how to design or implement something different. Proof of that is the answer from the MHCT officials when they were inquired about other technological alternatives for urban sanitation:*“You do not manage a city with onsite technologies. You need to manage wastewater. So, you require a service provider that guarantees that will mitigate these effects on the environment”.**“The ideal situation is, of course, a centralised solution because you need to conduct wastewater and take it to a place to treat it and comply with the environmental law”.**“In Colombia, onsite solutions are only implemented and suited for rural contexts”.*

The fragmentation of the WSS sector was also stated as the main reason for the siloed approach toward sanitation provision. The MHCT and the MESD constantly disagree as the former focuses on social development, and the latter controls and limits activities that might negatively impact the environment.

Lastly, participants constantly mentioned that the lack of technical capacity at a national level was a significant barrier to developing the sanitation regulatory framework. Three interviewees stated that officials in the MHTC do not know how to regulate CWIS as they are unfamiliar with technologies different from conventional ones. More specifically, participants pointed out that the Ministry officials lack the technical capacity to design regulations that deal with the communication and participation of the communities in sanitation projects as most of them have an engineering background.

## 4. Discussion

### 4.1. Does the Current Regulatory Framework Enable CWIS?

The Colombian regulatory framework does not explicitly forbids the implementation of CWIS but fails to create an enabling environment for it to flourish. A clear example is that few, if any, non-conventional projects are executed by the government, setting a lack of precedent and exposure to alternative sanitation approaches. Even within the *Agua al Barrio program* that is intended for informal settlements, a government official stated:

“*I am not familiar with those technologies* [referring to non-conventional alternatives], *within what I have managed in Agua al Barrio programme, which is the programme of differentiated schemes that is part of Decree 1272- 2017. I have not had the experience. The other types of individual solutions are more provisional and are things developed by the community itself to provide a solution for wastewater management, but…so far, we have not had a structured solution of this type*”.

Governmental reports recommend more resources to invest in the conventional approach to be able to expand coverage instead of proposing a shift in the strategy [47]. However, there has been some progress in shifting from the conventional to a more flexible approach to tackling urban sanitation. Law 1955-2019 and Decree 1688-2020 open the door to incorporating new technologies for contexts that require them. However, regulatory instruments need to be supported by the necessary laws and policies to be effective and sustainable. Without the amendment of Law 142-1992, which sets the framework for service delivery, the decrees and regulations that have been developed lack a legal framework to be implemented and prevent CWIS from being integrated into the system. Court ruling T 012-19 interprets Law 142-1994, widening the definition of public service by stating that it can be provided through a system of *physical or human networks,* allowing other approaches to be considered a public service. However, a court ruling does not replace a legal framework and does not provide a solid mandate to follow. As a result, several alternative service delivery projects have been blocked by some institutions from the WSS sector itself. It is therefore essential to update the legal framework to back the CWIS approach and define clear roles for the institutions in the sector to develop progressive regulatory instruments. 

As of now, households are left alone to solve their sanitation situation when they cannot be connected to the sewerage networks. Unfortunately, these solutions are not standardized, do not follow any technical guidance, and are not subject to a service delivery scheme. Additionally, key performance indicators have not been developed to hold the sanitation sector accountable. The only monitored indicator for the sanitation sector is the percentage of wastewater treated, and it only applies to conventional sewered solutions. Faecal Sludge Management is not mentioned within any policy or regulatory instruments. 

### 4.2. Nuances in the Results

While some stakeholders understand sanitation as an essential service that should be universal, others think of it as a service that can only be provided if certain basic conditions are given. Specifically, MHTC officials claimed that in some areas, there is nothing that the sector can do to solve issues determined by the context, such as illegal settlements, disorganised urban planning or lack of technological acceptance by the communities. They stressed that “*the difficulty ceases to be technical and becomes a social problem, which must be tackled before any sanitation solution can be implemented*”. 

Instead of adapting the regulations to the contexts, the MHCT stresses that the context is responsible for the poor sector results in the last 20 years of interventions. In this sense, the Ministry as the head of the sector should guide the policy to reach SDG 6.2 and 6.3. Nevertheless, there is no clear policy on integrating alternative technologies or service delivery models into the regulatory framework because regulatory responsibilities are divided between five different entities (MHCT, MESD, WRC, AEC and MARD). As a result, the implementation of new schemes and technologies are not initiatives proposed by the government, but rather case-based responses to the pressure of external entities such as development banks and cooperation agencies. 

Even if the country does not officially recognise CWIS as an approach to urban sanitation, some planning instruments released by the government acknowledged the importance of tailoring technologies to complex urban contexts [42]. An example of that is the Water and Sanitation Master Plan, which recognises that the government should incentivise innovative fit-to-context wastewater treatment technologies to reach the SDG targets in urban settings [48]. Nevertheless, different views exist within the government and regulatory agencies depending on the specific institution and the individual’s background. This research showed that Colombia does not have a unified multisectoral policy to tackle sanitation. The current approach is siloed with little coordination between the multiple agencies responsible for delivering safely managed sanitation. While some institutions, like the WRA, were actively engaged in developing a framework for CWIS, others did not even know what CWIS was and, once explained, thought of it as an absurd idea, unsuitable to Colombia’s reality. This revealed the sector’s lack of a coordinated approach and exposed sectorial silos that resulted in varying degrees of development on key parts of the regulatory framework.

### 4.3. What Is the Way Forward?

The fundamental areas to develop forCWIS identified by this study were the inclusion of differentiated service delivery models by updating the legal framework (Law 142-1994), the joint development of environmental regulations for a variety of technologies and contexts, the re-evaluation of land ordering requirements to access public services and the availability of subsidies to promote equity in sanitation. In order to pursue these developments, Colombia will need to build a comprehensive policy that incorporates strategic lines devised by a variety of players, not simply the MHCT as the sector’s leader. A clear policy with strategic lines about the urban sanitation subsector is crucial to tackling this issue in a comprehensive manner. Other countries’ experiences show that regulating and adopting CWIS is feasible. Rwanda, Tanzania, and Zambia have already begun to adjust their regulations for CWIS, and Colombia could use their experiences as a guide to developing its own legal framework [49,50,51].

Even if all these developments are necessary, experts have acknowledged that it is crucial to understand the broader political economy context to modify the regulatory framework [52]. This research shows that factors such as lack of political will and low capacity are determinants of poor reform outcomes [26]. As stated previously, the required reforms need the political will to coordinate the initiatives with other parties to get congressional approval. It is crucial to understand that Colombia is currently struggling with a pandemic, social turmoil after the protests of May 2021, and it is in a pre-electoral situation, so it is unlikely that this government will prioritise the reforms required to move CWIS forward. It is therefore important to consider the political economy related to sanitation to evaluate the feasibility of regulatory changes [53]. There is now a window of opportunity for reform to incorporate CWIS to the urban sanitation policy, as a new government will be elected in 2022 and the National Development Law for the following 4 years will be enacted at the beginning of the governmental period. 

On the other hand, it is critical to break the sectorial silos with an integrated approach and holistic planning. Effective coordination and unified planning can lead to positive interactions [11]. There is an area of opportunity in linking water, sanitation and waste management to offset the financial cost of sanitation provision and shift towards resource recovery [54]. Those developments have been driven by evolving the way sanitation is being thought of, by not only focusing on the toilet and the infrastructure but understanding that it involves the whole sanitation chain, from the collection and the transportation to the safe disposal and the use of valuable subproducts. 

Multilateral and external organisations have the opportunity to play a role in encouraging CWIS within the government, as shown by the Tumaco case study. They can act as exogenous drivers for reforms to promote the clear enactment of this public policy, sharing experiences from other countries, participating in international workshops, and including agreements to implement urban sanitation alternatives in the loan contracts [26]. To build capacity, development programs could also focus on strengthening the education institutions’ programmes to include the CWIS components to move apart from the conventional approach that continues to be taught today in universities and education centres in Colombia. 

## 5. Conclusions

This research demonstrates that the current regulatory framework in Colombia does not provide an enabling environment for CWIS. Law 142-1994 excludes onsite and shared sanitation technologies of the service delivery framework and leaves them without subsidies as compared to centralised sewered sanitation. Having a mandate at the top of the legislative hierarchy to amend this law is fundamental to facilitate the implementation of CWIS to direct regulation. In the meantime, court rulings have become an alternate way to allow the implementation of non-sewered sanitation. Even if they do not provide the legal standing that a regulatory framework requires, court rulings allow an expanded interpretation of existing laws while regulations develop, as seen in the case in Colombia.

The development of an appropriate regulatory framework for CWIS has been limited by other factors identified in this research. Political and institutional barriers comprise systemic resistance to change and strong adherence to conventional technology, hindering the robust legal adaptation needed to implement CWIS. Sectorial siloes have prevented sanitation from being tackled uniformly, and a state-wide policy must be developed to guide all the institutions to build regulations accordingly. In this regard, the development sector and international organisations have a significant role to play in sharing knowledge and advocating for new ways of conceiving sanitation. The Tumaco case study shows that change can be incentivised through such advocacy measures, funding opportunities, and pilot demonstrations. 

A comprehensive evaluation of Colombia’s regulatory framework helps understand the overall potential and barriers to implementing CWIS. The findings have shown that it is essential to regulate comprehensively and multi-dimensionally to execute this novel approach. That includes service delivery, land management, financial, environmental and technical regulations. Developments in one of the aforementioned aspects and not the others will not result in the effective and sustainable implementation of CWIS. It needs a well-coordinated approach from the national government, with a unified view on tackling urban sanitation and developing a holistic regulatory environment for its long-term success. Furthermore, based on the gaps identified, this paper gives specific recommendations for Colombia as well as general insights about the issues that can be encountered while implementing regulatory changes to incorporate CWIS. Colombia has a robust, well-funded WSS sector with some strong institutions. Thus, it has the potential to become a pioneer in Latin America for CWIS implementation. 

## Figures and Tables

**Figure 1 ijerph-19-05669-f001:**
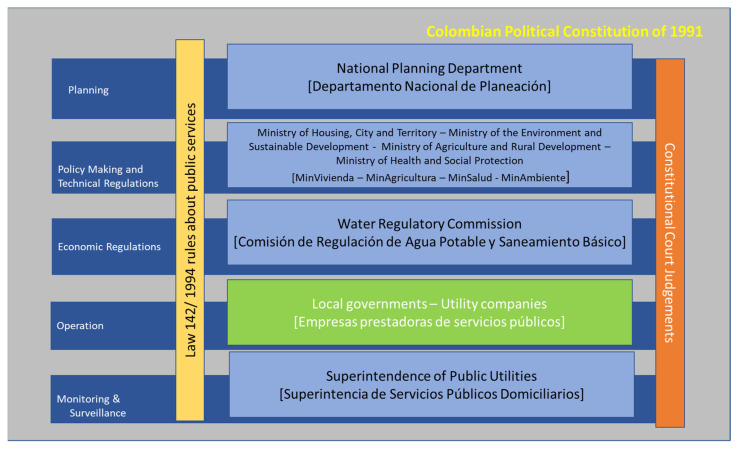
Key Stakeholders in the WSS sector.

**Figure 2 ijerph-19-05669-f002:**
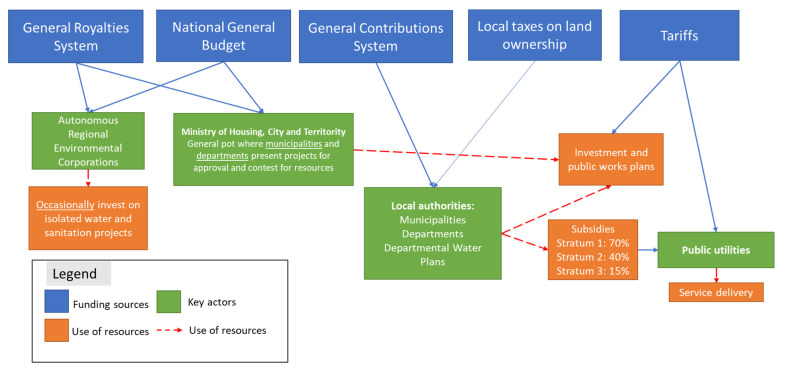
Financial sources in the WSS sector.

**Table 1 ijerph-19-05669-t001:** Type of key informants that participated in interviews and workshops.

Organisation	Number of Interviewees	Number of Participants in 4 Workshops	Scale
Ministry of Housing, City and Territory	3	6	National
Water Regulatory Commission	4	-	National
Superintendence of Public Utilities	1	-	National
Tumaco WASH Project Manager	2	-	Local
National Planning Department	1	3	National
Water and Sanitation Guild	2	-	National
Development Bank	1	2	National & Local
Academia	-	2	National
Independent WASH Consultant	1	-	National
Community-based Utility	1	-	Local
Local NGO	1	-	Local
Total	17	13	

**Table 2 ijerph-19-05669-t002:** Review of Colombian urban sanitation policies.

	Service Provision	Environmental	Land Management	Financial	Technical
**Constitution**	**Art. 49.** The State is responsible for environmental sanitation and granting every person access to public services **Art. 334.** The State must intervene progressively to ensure that the most vulnerable people have access to essential services **Art. 365.** Public services are inherent to the social purpose of the State that must ensure its efficient provision to all inhabitants of the national territory				
**Law**	**Law 142-1994: defines sanitation** as *the municipal collection of residues, primarily liquids, through pipes and channels, the complementary activities of transportation, and final disposal of such residues.* **Law 1753-2015:** defines a *differentiated scheme* for service provision and establishes three zones where these can be applied: zones of difficult access, complex management, or low connectivity. These schemes must comply with what is contained in the 142-1994 law.	**Law 2811-1974:** National Code of renewable resources and the protection of the environment.	**Law 388-1997:** LGs determine the territory’s distribution and organisation and establish a service area for WSS service provision. LG cannot provide public services in non-mitigable risk areas. **Law 136-1994:** states that LGs must deliver WSS services in the whole urban permitter **Law 2044-2020:** aims to solve the precarious situation of informal/illegal human settlements in urban contexts by instructing the municipalities to reach those settlements with services and solve their *informal* condition.	**Law 1176-2007:** Creates the GCS pot destined to be used by LGs for education, health, water and sanitation projects. Specifies that GCS investments must comply with what is contained in the 142-1994 law. (Art. 10-11).	**Law 1955-2019:** WSS solutions can be collective or individual under the figure of a *differentiated scheme*.
**Decree**	**Dec 302-2000 ^1^:** establishes the conditions to provide a public service. **Dec 1898-2016:** establishes an exception to Law 142 by allowing different technologies to be implemented. Nevertheless, *alternative technologies* do not fall under the service provision model and are excluded from subsidies. **Dec 1272-2017:** regulates service delivery in urban areas under special conditions that require a *differentiated scheme*. All the solutions must follow technical parameters contained in Res 0330-2017 **Dec 1688-2020:** non-conventional solutions are not subject to Law 142/94. Public utilities can operate *non-conventional* infrastructure only through a direct contract between the utility company and the user, which is not subject to subsidies. **Dec 1471-2021:** regulates conditions to connect buildings to water and sanitation utilities in urban areas	**Dec 1287-2014 ^2^:** regulates the use of sludge generated from wastewater treatment plants but does not include biosolids from other sources (onsite solutions) **Dec 1541-1978:** specifies that a land discharge permit must be obtained from the environmental authority to implement onsite sanitation solutions.		**Dec 1688-2020:** CGS resources can be used to finance studies and designs that include non-conventional solutions but not infrastructure.	**Dec 1688-2020:** states that onsite solutions can be applied in rural areas that require them, but does not mention urban areas
**Resolution**	**Res 688-2014:** defines a framework for WSS service delivery under normal conditions. **Res 949-2021:** defines a special service delivery framework for cases in which companies cannot comply with the efficiency parameters due to *particular conditions*. It includes a social factor in the tariff and allows for phased progress until the *particular conditions* are solved.	**Res 631-2015:** establishes requirements of land discharge permits for sewered sanitation **Res 699-2021:** establishes the requirements of the land discharge permit for onsite sanitation. **Res 1256-2021:** establishes requirements to allow wastewater re-use in agriculture		**Res 066-2019:** establishes the conditions to approve WSS projects contesting for national funds (GRS and NGB).	**Res 330-2017:** establishes technical parameters for WSS infrastructure. It details technical parameters for conventional technologies but briefly discusses non-conventional. **Res 779-2021:** changes technical requirements for conventional systems, condominial sewerage and onsite solutions. New technologies require approval from an accredited certification body.
**Court ruling**	T 012-19: Supreme court ruled that domestic public utilities are defined as those provided through a system of *physical or human networks* with terminal points at the users’ homes or workplaces and have the specific purpose of satisfying people’s needs.				

^1^ Dec 302/2000, Dec 1898/2016 and Dec 1688/2020 are now compiled in Dec 1077 /2015; ^2^ Dec 1287/2014–Dec 1541/ 1978 are now compiled in Dec 1076/2015.

**Table 3 ijerph-19-05669-t003:** GAP analysis of the regulatory arrangements in Colombia for urban sanitation throughout the sanitation service chain.

	Capture	Collection/Emptying	Collection/Transport	Treatment	Disposal/Re-Use
**Service** **Provider**	*Sewered systems and onsite** solutions*: house owners are responsible for building the facilities following technical requirements.	*Sewered systems*: utility companies (piped systems). *Onsite solutions*: contracted public utilities or private operators for emptying services.	*Sewered systems*: utility companies (piped systems). *Onsite solutions*: contracted public utilities or private operators for transport services.	*Sewered systems:* utility companies treat effluent in water treatment plants (Res 631/2015) and collect biosolids for re-use or disposal (Dec 1287-2014)	*Sewered systems:* utility companies treat and collect biosolids for re-use or disposal (Dec 1287-2014) coming from WTP
**GAP**	*Onsite solutions*: lack of technical standards and guidelines for onsite sanitation alternatives.	*Onsite solutions*: the service delivery model does not consider faecal sludge emptying	*Onsite solutions*: the service delivery model does not consider faecal sludge transport.	*Onsite solutions:* disposal of treated effluent in landfills is mandatory. So, faecal sludge is not treated.	Technical standards are too strict for small-scale utilities
**Financier of Facility**	*Sewered systems and onsite solutions**:* households in most of the cases. Sometimes, LG and AECs can support through neighbourhood/household improvement projects	*Sewered systems:* utility companies with local or national government financial support. *Onsite solutions:* utility companies or private operators	*Sewered systems:* utility companies with local or national government financial support. *Onsite solutions*: utility companies or private operators	Utility companies, the national government through MHCT or LG	Utility companies, the national government through MHCT or LG
**GAP**	*Onsite solutions:* limited financial resources for onsite sanitation programs.	*Onsite solutions*: GCS resources cannot be used for non-conventional sanitation technologies like vacuum tankers.		*Onsite solutions:* GCS cannot be used for the construction of non-conventional sanitation treatment technologies	
**Regulator**	Municipalities regulate technical parameters of household sanitary facilities, and AECs regulate effluent discharge for onsite sanitation	*Sewered systems*: SPU supervises the performance of utility companies.	*Sewered systems*: SPU oversees the performance of utility companies.	*Sewered systems:* SPU supervises the performance of utility companies and the correct application of tariffs. AECs regulate the effluent discharge	AECs regulate utility companies through environmental permits to centralised systems.
**GAP**	Regulations are not often enforced as AECs and LG lack the capacity.	Faecal sludge emptying is not regulated. Lack of guidelines and mechanisms to control private vacuum tankers	Faecal sludge transportation is not regulated. Lack of guidelines and mechanisms to control private vacuum tankers	Lack of guidelines for faecal sludge treatment processes. These are discharged into landfills without treatment	Faecal sludge re-use and disposal coming from non-sewered solutions are not regulated.
**Regulatory Instrument**	*Onsite solutions*: construction licenses required by municipalities and a discharge authorisation given by the AECs	*Sewered systems:* registering the utility company for SPU control	*Sewered systems:* registering the utility company for SPU control	*Sewered systems:* discharge permits (AECs) and discharge sanitation plan for decentralised and centralised WWTP.	Registering biosolids characteristics and quantity in the AECs
**GAP**	Discharge permits for onsite sanitation technologies are too strict	No regulatory instruments for vacuum tankers		No regulations for faecal sludge treatment from onsite solutions	Regulations only for faecal sludge coming from WWTP
**Payer of the Service**	Household	*Sewered systems: * tariffs, depending on the income level, can be partially subsidised. *Onsite solutions:* Household		*Sewered systems:* tariffs, depending on the income level, can be partially subsidised.	*Sewered systems**:* WSS service providers, municipalities in some cases.
**GAP**		No subsidies are available for faecal sludge emptying, transportation, treatment and disposal.			

**Table 4 ijerph-19-05669-t004:** Summary of the barriers to CWIS identified through the stakeholder consultation process.

Type	Barriers
General	- Law 142-1994 has a narrow definition of sanitation and favours the conventional approach to urban sanitation
Service Delivery	- The regulatory framework is limited to the technological options specified in Law 142, leaving out onsite and shared sanitation alternatives. - Disincentives to use technological alternatives which can only be used under special conditions. - Long and complicated regulatory instruments that are difficult to comprehend for small service providers. - Lack of incentives to expand to underserved zones outside of the official urban perimeter. - Faecal sludge emptying and transportation depend on unregulated agreements between house owners and private operators. They are not part of the service provision scheme. - Subsidies are only applicable to users of sewered sanitation.
Land Management	- Legality of expanding the coverage to areas that are not in the planning instruments of the municipalities is not clear. - Funding for sanitation projects require a certification of the land legality, posing a major barrier to providing services to informal settlements. - Public services cannot be provided in high-risk areas, but there is no alternative approach to provide a temporary solution while communities are moved.
Environmental	- Strict environmental permits even for small-scale sanitation alternatives that render projects unviable.
Financial	- Limited resources to fund non-conventional urban sanitation projects. - Social costs, essential for CWIS, to plan, select and adopt an adequate sanitation solution, cannot be included in the project’s budget.
Technical	- Technical guidelines focus primarily on conventional technologies

## Data Availability

The datasets generated for this study are available on request to the corresponding author.

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
