# Peer review of "Regulating Citywide Inclusive Sanitation (CWIS) in Colombia"

_ijerph, 2022, doi:10.3390/ijerph19095669_

Round 1

Reviewer 1 Report

In simple words, this study proposes regulations for CWIS in Colombia based on the identified challenges for running CWIS successfully.

Overall, the study is interesting and informative. It also fits well to the aimed special issue. Such practical activities can be very fruitful on our way toward achieving SDG6. 

The presented work shows the efforts of the authors. Nevertheless, such a high prestigious work deserves to be presented better. I feel there is room for improvement. Here are my comments:

1. Table 1 presents the characteristics of the community who participated in the interviews and workshops for discussions of challenges to achieve SDG6. This is fine. However, (to me) it seems that the end-users are neglected as we see no participant from the ordinary people or citizens. After all, the regulations will apply to people's lives, but from ordinary people, only 1 NGO has participated. How can regulations be defined while we do not know how people think about achieving SDG6? This should be addressed and discussed carefully.

2. The Same issue raises again in section 3.2. Understanding regulatory barriers through expert views: There is no discussion on social aspects of the framework. Do the experts have no idea on the acceptability, compatibility, and affordability of the frameworks in the community? I think it is worthy to at least discuss these important issues in section 4.3. and thinking about how to apply such frameworks sustainably.

3. Also, I invite the authors to discuss how they can evaluate the effectiveness of their framework for CWIS, specifically to provide sustainable sanitation for all. Several indexes are designed to evaluate the sustainability of a sanitation system in a community. Here is a list of some publications in this regard, sorted based on publishing time.
2020: https://doi.org/10.3390/su12176937
2015: https://doi.org/10.3390/su71114537
1999: https://doi.org/10.2166/wst.1999.0244

It will be great if the author may want to discuss these indexes in the introduction or discussion section of this paper and mention how applying such indices may be used for designing frameworks for CWIS or evaluating their applicability.

One minor comment: This work, may not be considered a classical scientific article. I feel it can be presented as a perspective if authors are willing to.

Overall, this study can be a fundamental one for processing several further studies. In particular, the presented GAP analysis of the regulatory arrangements in Colombia for urban sanitation throughout the sanitation service chain is a strength point of the work. I hope the authors find the above comments useful for revising their work.

Reviewer 2 Report

Overall comments

This is a clear, well-written and interesting paper on the regulatory context in Colombia regarding the provision of sanitation alternatives to sewers. The authors correctly point out that literature is lacking on this topic in Central and South America, and analyses on the regulatory environment will be welcomed by the sanitation sector which, to-date, has focused on technology but is increasingly appreciative of the political and enabling environment.

Most of my comments are fairly minor because I feel this paper is in a good state. My main critique is about how the authors frame and refer to CWIS. I’ve provided more detail on my thoughts in the specific comments below, but to summarise here I feel that the authors refer to CWIS to mean decentralised/non-sewered technologies when CWIS is actually a much broader paradigm. I think there is a need to check how CWIS is referred to throughout the paper, and a need for the authors to be a bit more transparent upfront about the specific focus of this study.

I was also hoping for a longer section on possible solutions/ways forward. The paper does a great job of outlining a multitude of obstacles to overcome, but sadly the proposed solutions feel limited. However, solving this problem is of course the hardest part and a challenge for the whole sector going forward, so it’s understandable the authors don’t have all the answers now.

Specific comments

Line 37: The number of significant figures here feels odd – it is probably fine to say 85.3% and 59.1%, or even 85% and 59%. Same with line 38.

Line 41: You might consider providing a reference here. The figure I’m used to hearing is the majority of the global population lives in “urban areas” rather than “cities” which aren’t entirely the same thing.

Lines 55-59: I think it might be worth articulating in more detail what the authors see as the components of CWIS for two reasons: One, the authors focus strongly on the non-sewered technological component of CWIS throughout this paper, to the point that they almost use CWIS as a synonym for decentralised technologies when I think CWIS is much more than that.

Second, there are different understandings of CWIS, so it would be good to know how the authors see it. The original CWIS principles included prioritising the human right to sanitation, addressing the entire service chain, integrating sanitation into the wider urban planning environment, and committing to work in partnerships to deliver sanitation. Since then, other authors have offered up more principles like funding for non-infrastructure aspects, political will and leadership, and providing incentives for good management.

Reforming regulations and supporting the use of multiple technologies are both components of CWIS, so I think it’s fine for the authors to focus on this. But I think they should acknowledge that CWIS is a framework/approach/set of principles that span many domains, and try to avoid using it synonymously with decentralised technologies.

Lines 68-69: Here is an example of where I think the framing of CWIS in this paper is a bit problematic. CWIS is more of a paradigm or (as the authors refer to it earlier) an approach. Speaking of it as something that can be implemented within an enabling environment reduces the CWIS concept. Many would say that CWIS is a primarily an approach to changing the enabling environment.

Lines 110-111: More specifically, I think this paper aims to analyse the Colombian regulatory environment for enabling the provision and management of decentralised technologies.

Lines 120-121: “Thus, a case study approach allows to go in-depth into the questions…” sounds a bit awkward. Consider re-phrasing.

Line 179: By “resources” do you mean financing/funding? Or does this go beyond financing?

Lines 201-204: This is an example of where I think CWIS is reduced to a synonym for decentralised or non-sewered technologies.

Lines 217-219: It might be useful to provide a few examples of these context-related difficulties.

Line 382: The PESTLE analysis wasn’t mentioned in the methods section – I think it should be added with a short description of what it is.

Line 428: Which Ministry is being referred to here?

Line 446: There are many good examples of barriers throughout the results section. I wonder if the authors may consider summarising them in a table at the end? This is not essential, but it might make the paper easer to reference for readers who do not have time to read the entire results section line-by-line.

Line 529: What do the authors mean by “closing the loop”?

Lines 557-559: My takeaway from reading the Tumaco case study was that the project was doomed to fail, but this sentence makes it sound like it was a success story?

Round 2

Reviewer 1 Report

The manuscript has been improved. Considering the authors' responses to the raised comments, this work can be published as an initial step. I wish to encourage the authors to continue conducting research in this field to cover knowledge gaps and weak points.

Author Response

Thank you very much for taking the time to review this paper.  It is very much appreciated.